# NeoCheck: A New Checklist to Assess Performance during Newborn Life Support—A Validation Study

**DOI:** 10.3390/children10061013

**Published:** 2023-06-04

**Authors:** Katharina Bibl, Felix Eibensteiner, Valentin Ritschl, Philipp Steinbauer, Angelika Berger, Monika Olischar, Vito Giordano, Michael Wagner

**Affiliations:** 1Division of Neonatology, Pediatric Intensive Care and Neuropediatrics, Department of Pediatrics, Comprehensive Center for Pediatrics, Medical University of Vienna, 1090 Vienna, Austriaangelika.berger@meduniwien.ac.at (A.B.); monika.olischar@meduniwien.ac.at (M.O.); michael.b.wagner@meduniwien.ac.at (M.W.); 2Department of Emergency Medicine, Medical University of Vienna, 1090 Vienna, Austria; felix.eibensteiner@meduniwien.ac.at; 3Institute of Outcomes Research, Center for Medical Data Science, Medical University of Vienna, 1090 Vienna, Austria

**Keywords:** assessment tool, Delphi process, newborn life support, resuscitation, simulation

## Abstract

Background: The aim of this study was to design and validate a new checklist and standardized scenario for assessing providers’ performance during Newborn Life Support (NLS). Methods: We invited twelve experts in Neonatology to take part in a three-step Delphi process. They rated the importance of each item of a newly designed assessment tool to evaluate participants’ performance during Newborn Life Support independently on a numeric rating scale from 1 to 5 (1 = lowest; 5 = highest) and were able to give additional comments. All items achieving a mean rating below four after the third round were deleted. For the reliability of the checklist, we calculated interrater reliability. Results: Using a standardized Delphi process, we revised the initial checklist according to the experts’ ratings and comments. The final assessment tool includes 38 items covering all relevant steps during NLS. The mean expert rating of all items was 4.40. Interrater reliability showed substantial agreement between the two raters in the first draft (κ = 0.80) as well as in the final draft of the checklist (κ = 0.73). Conclusion: We designed a feasible assessment tool for evaluating performance during NLS. We proved the checklist to be valid and reasonable using a Delphi validation process and calculating interrater reliability.

## 1. Introduction

Resuscitation of a newborn is a rare event with less than 0.3% of all newborn infants requiring chest compressions [1]. Therefore, the Newborn Life Support (NLS) algorithm needs to be regularly trained in simulation-based training settings in order to assure optimal care of critically ill infants, which is of utmost importance for the infants’ outcome [2,3]. In general, the NLS algorithm is internationally conducted according to either the European Resuscitation Guidelines (ERC) [1] or the guidelines of the American Heart Association (AHA)/Neonatal Resuscitation Program (NRP) [4]. Within the last decades, regular neonatal simulation trainings have been implemented in many institutions while training quality for Newborn Life Support courses markedly increased. Most instructors in the field of medical simulation complete high-quality train-the-trainer courses and are officially certified to teach NLS algorithms before conducting medical simulation training. This development ensures homogeneity and standardization of training programs, thereby enhancing the quality of training [5]. Furthermore, progress has been made in the implementation of new technical devices such as CPR feedback devices, Respiratory Function Monitoring, and video recording, which are used to analyze NLS performance [6,7,8]. This technical advancement effectively supports the feedback of traditional instructor-led training. However, it should be noted that the utilization of such devices requires financial resources to a certain extent. In most cases, the evaluation of participants’ overall performance is still carried out by instructors, which can be influenced by their perception and level of experience [9].

Standardized checklists to assess the performance of healthcare providers in a simulated or clinical NLS scenario are rare. In the field of neonatal resuscitation research most use the accordance of the participants’ actions with the Neonatal Resuscitation Program as the basis rating for performance [6,10]. A few case-based checklists have been developed for certain studies but did not undergo complete systematic validation processes [11]. The only checklists to assess neonatal resuscitation performance that has been shown to be both feasible and reliable tools and passed expert validation were published in 2005 and 2006 [12,13]. However, as guidelines have been constantly updated, these checklists are not applicable to assess performance in current NLS settings and are, therefore, outdated.

This study aimed to validate a checklist in accordance with official guidelines [1] to assess healthcare providers’ performance in Newborn Life Support with special emphasis on effective positive pressure ventilation. With this checklist and the corresponding simulation scenario, we aimed to provide a feasible tool for neonatal resuscitation research to allow objective assessment of the NLS performance of healthcare providers in a simulated or clinical setting.

## 2. Materials and Methods

### 2.1. Study Design

This study was conducted at the Pediatric Simulation Center of the Medical University of Vienna and aimed to design a checklist to assess the performance of healthcare providers during a simulated NLS scenario, which can also be translated and used in a real clinical setting. The first phase implied designing a relevant and realistic NLS scenario and a first draft of a corresponding checklist in accordance with official NLS guidelines. During the second phase, this first draft of the checklist was validated by international experts using a modified validation Delphi process. The reliability of the first draft as well as the final checklist was assessed by calculating inter-rater reliability (IRR). The study (Delphi progress) was given exempt status by the Institutional Review Board of the Medical University of Vienna in May 2020. For the used video material used for calculating Interrater Reliability (which was part of another study), approval was given by the Institutional Review Board and the Date Protection Committee of the Medical University of Vienna (EK Nr.: 1277/2018, July 2018).

### 2.2. Scenario Development

In the first step, two authors (KB, FE) designed a standardized and relevant NLS scenario with a focus on difficult airway management and the need for extensive resuscitation according to the NLS guidelines of the ERC 2015 [14], which have been the actual guidelines at the time of study conduction. In the scenario, health care providers needed to take care of an asphyxiated infant after an emergency Caesarean section in pregnancy week 40 + 4 due to acute maternal bleeding and suspected abruption of the placenta. Participants initially received a pre-briefing from our study team, followed by a preparation phase where they had the opportunity to set up the resuscitation unit and check all the equipment before the arrival of the newborn infant. Once the infant arrived at the resuscitation unit, participants were required to follow the ERC algorithm step by step, beginning with stimulating and warming the infant. Since there was no breathing effort observed, they had to initiate positive pressure ventilation using a flow-inflating bag. However, this method did not result in effective ventilation in any case. Therefore, participants had to employ several maneuvers to improve ventilation quality, such as using a guedel tube or switching to a different mask size. After implementing these actions, sufficient chest rise was observed during bag-mask ventilation.

However, despite these efforts, the infant’s heart rate remained below 60/min, necessitating the initiation of chest compressions. Additionally, consideration could be given to placing intravenous or intraosseous access, as well as administering fluid and adrenaline. After one minute of chest compressions, there was a return of spontaneous circulation. Details on the scenario (Figure A1).

### 2.3. Checklist Development

The development of the checklist was supported by intensive literature research with regard to existing checklists in newborn life support [12,13]. Based on the official NLS guidelines of the ERC and the given literature, especially by Lockyer et al. [12], the team established the first version of the case-based checklist, including 42 items in total. Thirty-five of those items followed a dichotomous character (done/not done), six items had three rating possibilities to choose from (not done/done incorrectly/done), and one item had four rating possibilities (not done/alternatives/done/done + alternatives).

### 2.4. Delphi Process

We asked 12 experts to participate in the Delphi validation process of our checklist. The reviewers represented an international panel from seven different countries (Canada, USA, Germany, Switzerland, Austria, Latvia, and Italy) and therefore guaranteed international expertise. All of them were pediatric consultants with specializations in Neonatology and/or Pediatric Intensive Care Medicine and were experts in Pediatric and Neonatal Simulation.

Experts were provided with a “best practice” video of the specific scenario to understand how the checklist items refer to certain parts of the scenario. The second part consisted of an online questionnaire (Survey Monkey Europe UC, Dublin, Ireland, www.surveymonkey.com, accessed on 14 April 2023) completed by every expert. To reduce the risk of bias, reviewers were blinded to each other throughout the study. They were requested to rate the importance of every item on a numeric rating scale (1 = lowest rating “not important” and 5 = highest rating “very important”). In case of a rating less or equal to three, they were asked to comment on their decision. which were then classified into one of the following categories: Spelling/Grammar, comprehensiveness, content, structure, redundancy, and limitation due to the simulation setting.

In the second round, every expert received an individualized document including the initial item, their given score, the mean score, their comment, and the modified item and were then asked to reevaluate their ratings. The rating system and request for further comments were similar to the first round of the Delphi process.

For the third round of the revision, we created a shorter document including only items with a mean rating of less or equal to four. This list consisted of the revised items, their individual score, the mean score, the anonymous comments of every reviewer, the suggestions for the final items, and the proposals for modification. A flow diagram of the Delphi process is depicted in Figure 1.

### 2.5. Reliability

To evaluate Interrater Reliability (IRR), independent experts in neonatology analyzed 63 pre-recorded videos presenting as many medical students participating in the designed NLS scenario. Therefore, the experts used the initial, non-validated draft of the checklist in the first step and completed the first draft of the checklist for every video independently. Videos with more than five missing items (e.g., missing data because of a lack of sound quality) were defined as non-usable. A total of 61 videos were analyzed for IRR. After the Delphi validation process, both experts repeated this procedure using the final and validated draft of the checklist. Thereby, experts also rated 61 videos independently.

### 2.6. Data Analysis

Categorical data were summarized using absolute and relative frequencies. The rating of the experts is reported as mean (SD) or median (IQR). For assessing the IRR of the checklist, we used weighted kappa analysis. All statistical analyses were performed using SPSS 24.0 (IBM Corp.; Armonk, NY, USA).

## 3. Results

### 3.1. Delphi Process

In total, ten experts completed all three rounds of the validation process. All had at least six years of clinical experience in either Neonatology or Pediatric Intensive Care medicine and were familiar with pediatric simulation training.

In the first round of the process, experts rated the importance of each item independently on a numeric rating scale from 1 to 5 (1 = not important; 5 = very important) and were able to give comments on every item. Eleven items (26.2%) had a mean rating below four. Details on the allocation to the given categories are shown in Table 1. After thoughtful discussion within the study team, we adapted items according to the reviewers’ suggestions. One item (oxygen supply) was initially removed due to a lack of evidence in the literature in 2015 (oxygen supply) but however, was re-attached when adapting to the 2021 guidelines.

Only four items (9.8%) did not reach a mean rating higher than four in the second review round. Half of the 41 comments in this round referred to spelling and grammar (51.2%), followed by comments on content (24.4%) (shown in Table 1). After the third round, three items still did not reach a mean rating above four and, therefore, were removed from the checklist. One item was removed due to redundancy (shown in Table 2).

Initially, 37 items met the criteria for consensus and were included in the final checklist, with 30 items being dichotomous and seven providing more than two categories (e.g., done/insufficiently done/not done), and a maximum total score of 46 points. The mean expert rating of all items on the final checklist was 4.40 on the numeric rating scale. After the release of the new guidelines 2021, we added one question (“Did the participant apply 100% oxygen?”, see discussion section) and, therefore, ultimately the checklist included 38 items with a maximum total score of 48, divided into a preparation part (10 items) and a resuscitation part (28 items) (Figure A2).

### 3.2. Reliability

To assess the checklist’s reliability, two independent study team members evaluated 61 recorded videos, in which medical students performed in the given NLS simulation scenario. The interrater reliability for the first draft of the checklist showed a score of κ = 0.80 (*p* < 0.05) while the IRR of the final checklist was κ = 0.73 (*p* < 0.05) (shown in Figure 2).

## 4. Discussion

As extensive resuscitation of a newborn infant is extremely rare [1], regular training sessions are of high importance to assure adequate knowledge and skills within medical staff in case of neonatal emergencies [2,3]. To assess health care providers’ performance in training but also in real clinical settings, valid and reliable assessment tools are essential to reduce any bias by trainer expertise or teaching styles. Our study team created a new assessment tool to evaluate healthcare providers’ performance during a standardized NLS scenario. We, therefore, designed a relevant NLS scenario and corresponding assessment checklist, including a preparation part. The validity of this tool was established through a three-round Delphi process involving ten field experts who provided independent feedback.

The final version of the checklist comprises 38 items, with a maximum possible score of 46 points. The mean expert rating for all items was 4.40, indicating a favorable assessment of the tool’s content and relevance. Furthermore, the interrater reliability analysis demonstrated substantial agreement between the two raters during the final drafting stage of the checklist, with a kappa value of 0.73.

### 4.1. Validity

We used the Delphi validation method as a standardized and broadly accepted tool to validate a newly developed assessment tool. A Delphi study is an ideal means to assess content validity defined as the degree to which the content of an instrument is an adequate reflection of the construct to be measured [15]. Experts are blinded to each other to reduce any bias by other experts’ opinions. This feasible process has been used in several settings to assure high quality content within checklists and scenarios [16,17,18,19,20,21,22,23]. Our study showed high validity with a total score of 4.4 (ranging from 1–5). Banayan et al. [20] used a lower cut off value of three out of five possible rating points. Furthermore, Johnston et al. [18] set a cut off value of four out of seven possible ratings, which was also set lower than ours. Our defined cut off value of 80% (at least 4 out of 5 points) was placed above the ones used in previous studies as strong consensus is essential in the field of official guidelines. Our developing process of the checklist was similar to the process of Schmutz et al. [19]. The main differences were (i) that they have chosen fewer experts for the validation process (*n* = 5) (ii) that we did not do a pilot testing but compared 61 recorded videos with the initial and the revised checklist. The recommended amount of selected experts for the Delphi process varies from 5–30 experts [24]. Therefore, our selection of experts (*n* = 10) should be in an adequate range. In previously published Delphi studies, the number of reviewers ranged from 5–12, which seemed to be acceptable to provide sufficient validity [17,18,20]. Therefore, our method reduced opportunities for subjective interpretations and thus minimized rater biases.

### 4.2. Reliability

The Interrater Reliability describes the agreement between raters using the same tool or examining the same data [25]. Interrater Reliability during the assessment of the initial checklist was κ = 0.80 (*p* < 0.05), while the IRR of the final checklist was κ = 0.73 (*p* < 0.05) (Figure 2). According to Landis et al., these scores show substantial agreement (κ = 0.61–0.80) between the two raters in both versions of the checklist [26]. In comparison to van der Heide et al. [13] with an IRR of κ = 0.77 and Johnston et al. [18] with an IRR of κ = 0.79, our results showed similar data. Even though we could not reach a perfect IRR (κ = 0.81–1.0), our checklist is a reliable tool for NLS assessment.

### 4.3. The Checklist

This is not the first checklist on NLS performance but represents the most valid, reliable, and recent one. Our final checklist includes 38 items, comparable to other checklists in the field [12,13]. The NLS assessment tool from Van der Heide et al., including 44 dichotomous items, represents a valuable instrument for assessing neonatal resuscitation skills in a training setting showing good inter- and reasonable intra-rater reliability [13]. In contrast to our study, skills are not assessed chronologically as they appear in the scenario but are grouped into eight different subunits (group function, preparation and initial steps, communication of heart rate to lead resuscitator, oxygen administration, bag/mask ventilation, intubation, chest compressions, and drugs). This approach seems reasonable but makes the evaluation of complex NLS scenarios more difficult as there is no chronological order to follow. Their checklist covers all important steps of NLS management, including intubation and drug administration. On the other hand, we narrowed it down to the initial NLS management with a focus on preparation, temperature management, sufficient bag-mask ventilation, and high-quality chest compressions as these remain the most important steps to improve patients’ survival [1]. We believe that an assessment tool that is overly complex and includes an excessive number of items may hinder its practicality and reduce its usage in the long term.

Lockyer et al. presented a feasible and practicable tool for NLS assessment, following NRP guidelines [12]. This checklist consists of 22 items. It is supposed to meet the criteria for Megacode Skills and, therefore, needs to be compact and easy to follow during the live evaluation of simulation training. As we designed our checklist following this tool, content is comparable in most items. However, we extended some items to manage the difficult airway when bag-mask ventilation is not sufficient, as adequate ventilation is the key factor in the stabilization of the newborn child. Furthermore, we separated some items into individual ones to have clear and precise items without any room for misinterpretation, especially in the preparation part. Therefore, it is safe to say that existing NLS assessment tools represent reasonable checklists for teaching purposes. However, those have not been validated via a standardized Delphi method and are outdated as published in 2005 and 2006.

There are several items we did not include in the initial checklist on purpose. First, we did not include delayed cord clamping as in many NLS simulations it is not part of the training content, even though it remains an essential step in a real-life delivery [1,27]. However, if settings allow for the simulation of late cord clamping, it could certainly be incorporated into such assessment tools. This would be particularly relevant for interdisciplinary NLS trainings involving Obstetricians and Neonatologists, enhancing the overall training experience.

Furthermore, we excluded the item “Did the participant apply additional oxygen?” as this was not supported by ERC guidelines 2015. However, the new guidelines do recommend applying 100% oxygen in case of chest compressions [1]. We also excluded the item concerning assessing the baby’s color, as we know from recent literature that its interpretation is rather vague and does not give qualitative information about the infant’s condition [28]. The scenario itself was newly developed for another study, where external stress parameters should distract participants and increase stress levels. Therefore, in the scenario, the nurse leaves the room after the baby’s initial stabilization. During the Delphi process of the checklist, reviewers questioned the importance and empirical realism of this aspect for a general NLS scenario. Furthermore, the initiation of fluid did not meet the reviewers’ requirements for a general NLS setting. Therefore, we deleted these items.

We discussed the inclusion of a timeframe for all the steps during NLS extensively. However, we distanced ourselves from defining timelines as determining the most appropriate timing for many items, such as when to change wet towels, is challenging. The guidelines do not provide specific timeframes for such actions, as it largely depends on the situation and the individual baby. However, we recognize that this step is crucial in the course of resuscitation.

In the end, we distanced ourselves from defining a grading system according to the score reached during the assessment. Frankly, items would have to be weighted in terms of their importance for the infant’s survival first, before valuable grades could be given to the providers (e.g., sufficient ventilation is more important than wearing gloves).

### 4.4. Adaptation to 2021 ERC NLS Guidelines

The checklist and corresponding scenario were developed and validated in 2020 and therefore followed recommendations from the ERC NLS guidelines 2015 [14]. However, a new guideline was published in April 2021, showing minor alterations to the previous ones [1]. To summarize the amendments, there is more focus on airway management, including the early use of alternative airway devices such as a laryngeal mask and prolonged initial bag-mask ventilation (30 s instead of a second set of five ventilations). As the main interest of the study team is in providing a feasible and updated checklist, we decided to revise the checklist and corresponding scenario so that they meet all criteria from the 2021 ERC guidelines. Therefore, we changed question nr. 26 from “Did the participant perform another 5 ventilations after having assessed the low heart rate?” to “Did the participant continue positive pressure inflations for another 30 s after having assessed the low heart rate?”. Furthermore, we changed question Nr. 28 from “Did the participant assess heart rate after another 5 ventilations” to “Did the participant assess heart rate after 30 s of effective inflations?”. Moreover, we added question Nr. 32 “Did the participant apply 100% oxygen during chest compressions?”, as this is clearly recommended in the 2021 guidelines. These changes are essential to provide an NLS assessment tool following the latest recommendations. However, the above-mentioned amendments were not validated within the Delphi Process of the original checklist, as the process had already been completed by the time the 2021 ERC NLS guidelines were published. In the final checklist (Figure A2), all alterations that have not been validated during the Delphi Process are highlighted with an Asterix.

### 4.5. Limitations

Our study has several limitations. First, two authors (KB, FE) designed the checklist and assessed the previously recorded videos to evaluate IRR. In general, independent reviewers of the videos should be favored. Finally, we did not include any assessment of communication, leadership, or teamwork in our checklist, although human factors are a significant source of error during resuscitation [29,30]. However, this is a complex task and has to be taken into consideration in future checklist developments.

## 5. Conclusions

We developed a feasible and intuitive tool to assess healthcare providers’ performance in neonatal resuscitation that shows good validity and high reliability. The checklist and corresponding scenario proposed in our study can serve as a foundation for various simulation and resuscitation studies in the future. Moreover, they can be readily adapted for implementation in clinical settings, including those in low-resource countries. In the next step, the assessment of human factors and teamwork as well as the use of objective feedback devices should be added to the assessment tool.

## Figures and Tables

**Figure 1 children-10-01013-f001:**
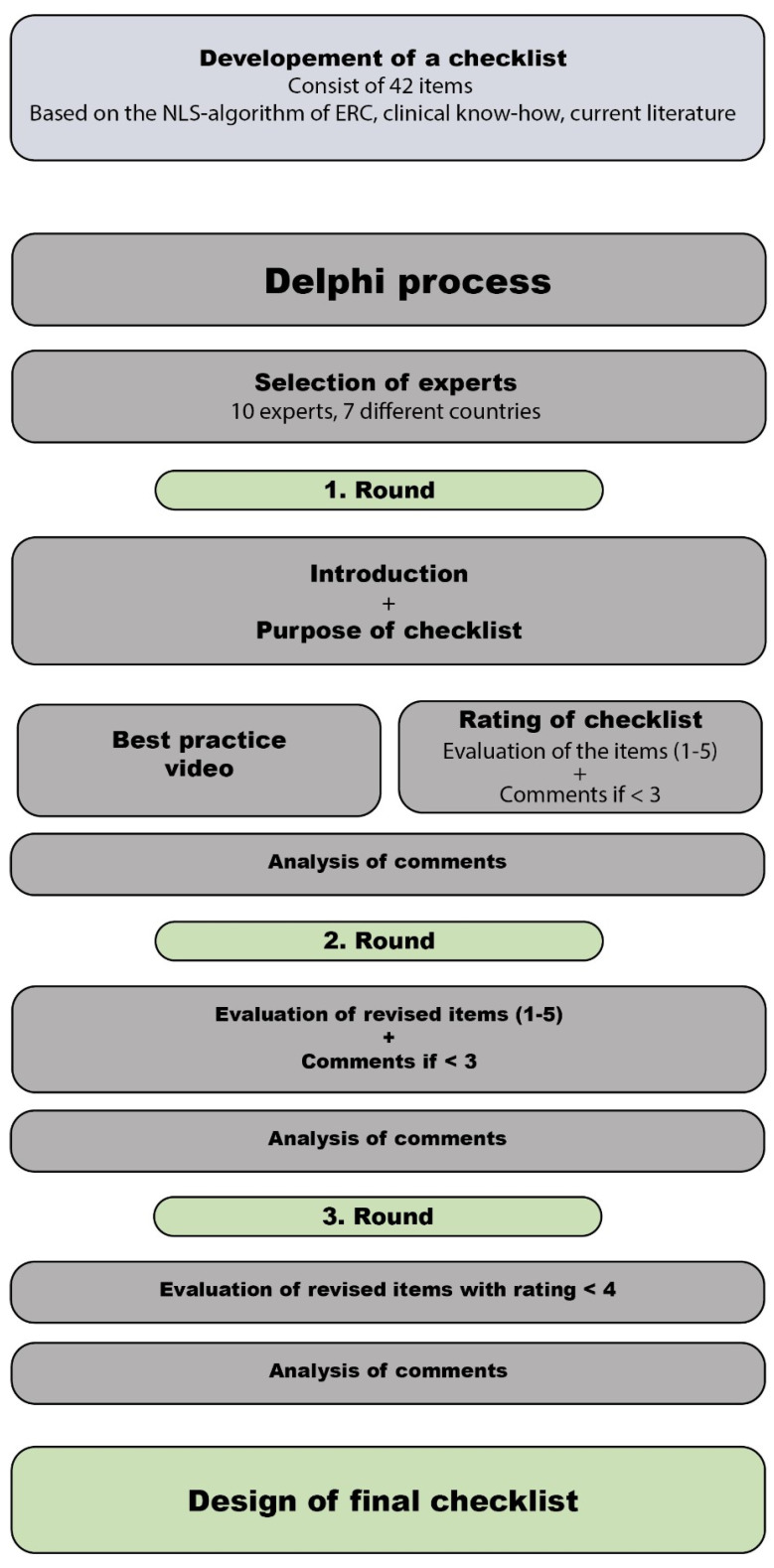
Flowchart of the Delphi process.

**Figure 2 children-10-01013-f002:**
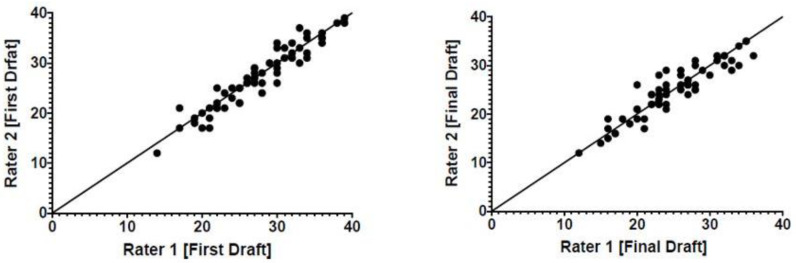
Interrater Reliability (IRR) of first draft of the checklist ((**left**); κ = 0.80) and final version ((**right**); κ = 0.73) between two experts.

**Table 1 children-10-01013-t001:** Reviewers’ comments categorized.

	Number of Comments Review 1	Number of Comments Review 2
Limitation due to simulation setting	8 (7.7%)	6 (15.0%)
Language/Grammar	35 (33.7%)	21 (51.2%)
Comprehensiveness	11 (10.6%)	1 (2.4%)
Content	36 (34.6%)	10 (24.4%)
Structure	2 (1.9%)	2 (4.9%)
Redundancy	12 (11.5%)	1 (2.4%)
Total	104	41

**Table 2 children-10-01013-t002:** Items removed from the checklist during the Delphi Process.

Question	Mean Rating	Reason for Elimination
Did the participant apply additional oxygen?	4.10	Lack of evidence in the current literature
Did the participant assess the baby’s colour?	3.35	Rating < 4, lack of importance in NLS, limitations in a simulated setting
Did the participant initiate to administer fluid?	3.90	Rating < 4, given the history of abruption of the placenta, may be important, but not in a general NLS scenario
Did the participant give verbal concerns about the decreasing heart rate?	4.35	Limitations due to simulation setting, the nurse leaving the room somewhat unrealistic
Did the participant call the nurse back to the resuscitation unit?	3.90	Rating < 4, Limitations due to simulation setting, the nurse leaving the room somewhat unrealistic

## Data Availability

The data that support the findings of this study are available from the corresponding author K.B., upon reasonable request.

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
