# Peer review of "NeoCheck: A New Checklist to Assess Performance during Newborn Life Support—A Validation Study"

_children, 2023, doi:10.3390/children10061013_

Round 1
Reviewer 1 Report
The resuscitation maneuvers can also be evaluated by computers connected to a self-inflating bag. These types of computers will provide a diagram that helps improve ventilation skills
So not only the instructors can evaluate the resuscitation steps.
Another method used in the past was filming the resuscitation scenarios and evaluating the maneuvers recorded for each team.
Any kind of evaluation that will lead to an improvement in the resuscitation maneuvers is welcome. So your evaluation technique should be a good step in NLS programs.
The English is good enough
Author Response
Dear Reviewer,
Thank you very much for your valuable comments and suggestions. The study team highly appreciates your expertise in the field of neonatal resuscitation. We agree that there are several strategies available for providing feedback in resuscitation and ventilation training. Respiratory Function Monitoring and video recordings with debriefings are increasingly being utilized in neonatal resuscitation settings and simulation trainings. The incorporation of new technologies, such as augmented and virtual reality, is now commonplace in many of our current simulation studies and should definitely be considered for implementation in future trainings, particularly in high-resource simulation centers. We mentioned this in the introduction part (line 39).
However, it is important to note that our study team specifically aimed to create a checklist that is universally applicable, including in low-resource countries. Therefore, we intentionally designed this assessment tool without incorporating recent technological advancements. This approach allows for broader accessibility and usability across various settings. Nonetheless, we acknowledge the significance of these new technologies and their potential benefits in enhancing training and improving outcomes.
Furthermore, we would like to emphasize that the checklist and corresponding scenario proposed in our study can serve as a foundation for further feasibility testing in the future. This opens up opportunities for incorporating additional elements, such as respiratory function monitoring and video recordings, in subsequent iterations of the assessment tool.
Once again, we sincerely appreciate your insightful feedback.
Reviewer 2 Report
I thank the authors for the opportunity to have read this interesting paper.
I congratulate you for your contribution to the evaluation of resuscitation management in newborns.
Author Response
Dear Reviewer,
Thank you for your feedback and kind words regarding our paper. We greatly appreciate the opportunity to share our work and contribute to the evaluation of resuscitation management in newborns. Your words encourage and motivate us to further delve into this important area of research.
We sincerely thank you for taking the time to review our paper and for your positive feedback.
Reviewer 3 Report
1. Please describe the summary and significance of this study at the beginning of the discussion.
2. It is difficult to see the contents clearly due to the low resolution of the picture (Figures). Please increase the resolution of the picture (Figures).
The rest of the content is well organized, so there is nothing special to edit.
Author Response
Dear Reviewer,
Thank you very much for providing us with your valuable feedback and comments on our manuscript. The study team greatly appreciates your expertise in the field and your suggestions to enhance our work. We would like to address your suggestions point by point:
- In response to your suggestion, we have included a summary of the study results and aims at the beginning of the discussion section (lines 178-193).
- We have taken your feedback into consideration and made improvements to the figures. Specifically, we have enhanced the resolution of the figures and revamped the layout of the final checklist, changing it from a panel format to a landscape format.
Once again, we sincerely thank you for your insightful comments and suggestions.
Round 2
Reviewer 1 Report
Thank you for your response and for the modified version of the paper.
In my opinion, the revised version of the paper meets the requirements for publication.
The English is fine....no need for further corrections
Author Response
Dear Reviewer,
thank you very much for your positive response. We appreciate your time and effort on this work.